# Ligustrazine as an Extract from Medicinal and Edible Plant Chuanxiong Encapsulated in Liposome–Hydrogel Exerting Antioxidant Effect on Preventing Skin Photoaging

**DOI:** 10.3390/polym14214778

**Published:** 2022-11-07

**Authors:** Chang Liu, Ying Xia, Yufan Li, Yongfeng Cheng, Hongmei Xia, Yu Wang, Yan Yue, Yifang Wu, Xiaoman Cheng, Yinxiang Xu, Zili Xie

**Affiliations:** 1College of Pharmacy, Anhui University of Chinese Medicine, No. 350, Long Zi Hu Road, Hefei 230012, China; 2Clinical College of Anhui Medical University, Hefei 230031, China; 3School of Life Science, University of Science and Technology of China, Hefei 230027, China; 4Anhui Province Key Laboratory of Pharmaceutical Preparation Technology and Application, Hefei 230012, China; 5Zhaoke (Hefei) Pharmaceutical Co., Ltd., Hefei 230088, China; 6Anhui Institute for Food and Drug Control, Hefei 230051, China

**Keywords:** ligustrazine hydrochloride, liposome–hydrogel, in vitro and ex vivo release, antioxidant

## Abstract

Long-term sunlight exposure will cause the accumulation of free radicals in the skin and lead to oxidative damage and aging, antioxidant drugs have gradually become the focus of research, but there is little research on antioxidant drugs for percutaneous treatment. The purpose of this study was to prepare ligustrazine hydrochloride (TMPZ)-loaded liposome–hydrogel (TMPZ-LG), evaluate its antioxidant properties, and apply it on the skin of mice to observe whether it had preventive and therapeutic effect on the irradiation under the ultraviolet rays, in an attempt to make it into a new kind of delivery through the skin. TMPZ-LG was prepared by the combination of film dispersion and sodium carboxymethylcellulose (2%, CMC-Na) natural swelling method. The release rates in vitro permeation across the dialysis membrane and ex vivo transdermal had both reached 40%; the scavenging effect of TMPZ-LG on 1,1-diphenyl-2-picrylhydrazyl (DPPH) and H_2_O_2_ were 65.57 ± 4.13% and 73.06 ± 5.65%; the inhibition rate of TMPZ-LG on malondialdehyde (MDA) production in liver homogenate and anti-low density lipoprotein (LDL) oxidation experiments ex vivo were 15.03 ± 0.9% and 21.57 ± 1.2%. Compared with untreated mice, the skin pathological symptoms of mice coated with TMPZ-LG were significantly reduced after ultraviolet irradiation, and there was statistical significance. The results showed TMPZ-LG could exert good antioxidant activity in vitro and ex vivo; therefore, it is feasible to prevent and treat skin oxidation.

## 1. Introduction

With the improvement in people’s quality of life, people are paying increasingly more attention to oxidation damage and aging. The skin is the largest organ of the human body, accounting for 10–15% of the body weight [1]. Once oxidative damage occurs, it will lead to skin aging and other diseases. Photoaging accounts for more than 80% of facial aging [2]. The concept of antioxidation is not only applied to clinical treatment of diseases but also widely used in the field of cosmetics. Antioxidation and antiaging of skin are the focus of skin care.

Ultraviolet rays (UV) are the main cause of oxidation and aging of skin, which are divided into ultraviolet A (UVA) (315–400 nm), ultraviolet B (UVB) (280–315 nm) and ultraviolet C (UVC) (100–280 nm) [3]; among them, UVA and UVB can induce oxidative aging of human skin. UVA can reach the dermal layer to denature the collagen in the skin and weaken the skin elasticity [4], UVB could reach the epidermis and cause erythema and wrinkles on the skin [5]. This pathological reaction is mainly caused by the production of reactive oxygen species (ROS) [6]. In the skin, about 2% of oxygen consumption is converted into ROS through internal processes [1]; once the ROS accumulate, an inflammatory reaction would occur, thus accelerating the oxidative aging of skin, so it is very important to resist ROS to exert antioxidation. 

Chuanxiong (CX) is a traditional Chinese medicine plant, which can be used as food and medicine, and the rhizome of CX is the main edible and medicinal part. It is a typical therapy to keep healthy by consuming the rhizome with food. For example, stewed CX with duck meat can promote blood circulation and remove blood stasis; CX wine can be used for treating joint pain and dysmenorrhea in women. Ligustrazine is one of the effective components of CX, and its hydrochloride form is often used for clinical treatment. Ligustrazine hydrochloride, also known as tetramethylpyrazine hydrochloride (TMPZ), not only has the function of activating blood circulation and removing blood stasis, but also has strong antioxidant activity. Studies have shown that TMPZ can not only scavenge ROS [7], but also inhibit the epidermal and dermal cells secreting inflammatory factors in large quantities, such as interleukin (IL), cyclooxygenase-2 (COX-2) and tumor necrosis factor (TNF-α) [8]. In addition, TMPZ can also prevent the excessive production of melanin of the skin by inhibiting the activity of tyrosinase.

Therefore, based on the antioxidant properties of TMPZ, we prepared it into TMPZ-LG, which is a novel preparation as the transdermal delivery. In recent years, the transdermal delivery system has become a hot topic which acts on the skin surface, absorbs drugs through capillaries and reaches the blood circulation of the whole body, thus exerting local or systemic effects [9]. Compared with other drug delivery routes, such as oral administration and injection, transdermal drug delivery is convenient, which could avoid the first-pass effect of liver and the pH effect of gastrointestinal tract, and has the effect of sustained and controlled release. The liposome–hydrogel [10] delivery system can not only exert the characteristics of the sustained and controlled release of liposomes [11,12,13] and stimulation-responsive release of the drug [14], but also improve the stability of liposomes in the hydrogel three-dimensional space network structure [15], and better adhere to the surface of the skin. 

## 2. Materials and Methods

### 2.1. Materials

TMPZ (purity 99%), soybean phospholipids (purity 99%) and cholesterol (purity 99%) were purchased from Shanghai Jinsong Industry Corp., Ltd (Shanghai, China). Carboxymethylcellulose sodium (CMC-Na) was purchased from Tianjin Guangfu Institute of Fine Chemical Industry. DPPH and 2-thiobarbituric acid (TBA) were obtained from Shanghai yuanye biology science and technology Corp., Ltd (Shanghai, China). FeSO_4_∙7H_2_O was purchased from Sinopharm Chemical Reagents Co., Ltd. (Shanghai, China). Trichloroacetic acid (TCA) was obtained from Damao Chemical Reagent Factory (Tianjin, China). T-SOD Test kit was acquired from Jiancheng Bioengineering Institute (Nanjing, China). All chemical reagents were of analytical grade, and deionized distilled water was used in all experiments.

### 2.2. Animals

Healthy Kuming female mice (22 ± 2 g) were purchased from the Animal Experimental Center of Anhui University of Chinese Medicine (Hefei, China). All animal experiments were carried out according to the guidelines approved by the ethics committee of Anhui University of Chinese Medicine (Hefei, China).

### 2.3. Preparation of TMPZ Solution

Adopting the preparation method of El-Badry et al. [16], briefly, 0.06 g powder of TMPZ was weighed accurately, placed in a 100 mL brown volumetric flask, the volume was then set to the scale with phosphate buffer solution (PBS, pH 7.4), shaken well to obtain a TMPZ solution with a concentration of 0.6 mg/mL and stored in the dark at 4 °C for later use.

### 2.4. Preparation of TMPZ Loaded-Liposomes (TMPZ-L)

We prepared TMPZ-L by film dispersion method [17,18] and chose the mass ratio of phospholipid to cholesterol to be 3:1. Amounts of 0.60 g phospholipid and 0.20 g cholesterol were weighed accurately, and 10 mL absolute ethyl alcohol was added into a round-bottom flask, which was completely dissolved in an ultrasonic water bath (70 °C) instrument. While the bottom flask was rotated, the ethanol was slowly evaporated, and then under the action of the rotary evaporator, an adhesive liquid film formed on the flask wall. A total of 10 mL of TMPZ (0.6 mg/mL) was placed into the film-forming flask and ultrasonic mixing until it was completely hydrated, then it was filtered across the microporous membrane with pore diameter of 0.22 μm three times to obtain TMPZ-L. The above steps were followed replacing the TMPZ solution with PBS to prepare blank liposomes (B-L). 

The zeta potential and particle size of TMPZ-L were measured by using a particle size analyzer (Malvern Instruments Ltd., Malvern, UK) at 25 °C with a 90° scattering angle [19]. The sample was diluted 5-fold with PBS and transferred to a specific cell for analysis. The values are means of triplicate analyses.

A quantity of 1.0 mL of TMPZ-L was added to a 10 mL volumetric flask, and the volume was set to the scale with PBS, a 4 mL sample was removed, then centrifuged at 4500 revolutions per minute (rpm) for 15 min, the supernatant liquid was then obtained and the absorbance at the wavelength of 295 nm was measured and the concentration of drug which was not encapsulated into liposomes could be calculated according to the standard curve. Another 1.0 mL of TMPZ-L was removed and placed in a 10 mL volumetric flask, ethanol was added to a constant volume, ultrasonically dissolved liposome membrane, the supernatant liquid was then removed, followed by the same steps as above, and the whole concentration of drug could be calculated according to the standard curve, the encapsulation rate (ER%) of TMPZ-L was calculated by equation:
(1)$$\mathrm{ER}\%=\frac{\left[C_{w}-C_{n}\right]}{C_{w}}*100\%$$
where “Cn” was free drug concentration in liposomes; “Cw” was whole drug concentration in liposomes.

### 2.5. Preparation of TMPZ-Loaded Hydrogel (TMPZ-G)

The TMPZ-G was prepared according to the description of El-Badry et al. [16] and simply modified. Briefly, 0.1, 0.2, 0.3, 0.4, and 0.50 g CMC-Na powder was weighed in a 10 mL test tube, the volume was fixed with TMPZ solution (0.6 mg/mL) and stirred quickly. After the powder was almost completely dissolved, it was left to stand for 24 h to obtain hydrogel with a different percentage (1%, 2%, 3%, 4% and 5%) and the best percentage was chosen to prepare the final TMPZ-G.

### 2.6. Preparation of TMPZ-Loaded Liposome–Hydrogel (TMPZ-LG)

A solution of 10 mL of TMPZ-L and B-L was prepared by film dispersion method according to the above steps, and then sprinkled with 0.20 g of CMC-Na, stirred quickly until all particles were dissolved and left to stand for 24 h and become completely swollen; thus, TMPZ-LG and B-LG were obtained.

A quantity of 0.10 g of TMPZ-LG was obtained, and 3 groups were set up in total, particles were dissolved in 5 mL distilled water by ultrasound and the pH value was measured when they were fully dissolved. According to the standard of Chinese Pharmacopoeia, the pH value of liposome–hydrogel should generally be checked, and it was necessary to test whether it was irritating to the skin.

The stability of TMPZ-LG was determined by accelerated centrifugation. An amount of 6.0 g of TMPZ-LG was placed in a centrifuge tube, and set in 3 groups, centrifuged at 6000 rpm for 15 min, then it was removed to observe whether there was stratification.

### 2.7. In Vitro and Ex Vivo Release Rate of TMPZ, TMPZ-L, TMPZ-G and TMPZ-LG across Dialysis Membrane and Mice Skin

The release rate (R%) of the drug was investigated by Franz diffusion pool (Figure 1). As an in vitro simulation device, the Franz diffusion cell could be effectively used to study the in vitro release characteristics of transdermal preparation. The constant-temperature circulating water kept the constant-temperature working state of the diffusion cell, and constant-speed magnetic stirring ensured the solution was evenly distributed. 

The volume of the diffusion cell was measured and PBS was used to fill the receiving pool and place it into a small stirrer, spread the dialysis membrane onto the receiving pool and ensure there were no bubbles between the supplying pool and the receiving pool. The Franz diffusion cells were placed on a constant temperature magnetic stirrer, and the temperature was set at 37 °C. An amount of 1.0 mL of PBS, TMPZ, BL, TMPZ-L, B-G, TMPZ-G, B-LG and TMPZ-LG was into the supplying pool and 2 mL of the samples from the receiving pool were removed at time points of 5 min, 10 min, 20 min, 30 min, 60 min, 120 min, 2 h, 3 h, 4 h, 5 h, 6 h, 7 h, 8 h, 9 h, 10 h, 11 h, 12 h, 24 h, 36 h, 48 h, 60 h, 72 h and 84 h. When every sampling was completed, 2 mL of PBS was injected into the receiving pool. 

The skin on the back of mice was used for the transdermal release of the drug. The hair on the back of mice was shaved and 0.2 mL of 20% urethane per 20 grams of body weight was injected into it, the skin was peeled with scissors and the excess subcutaneous fat was removed with tweezers. The next operation steps were the same as the dialysis membrane in vitro except the skin of the mouse replaced the dialysis membrane. The absorbance of the samples was measured at the wavelength of 295 nm and the R% was plotted according to the following equation: (2)$$A=0.0382*C-0{.0079\;(R}^{2}=0.9994)$$
(3)$$R\%=\frac{C_{n}{*V}_{n}+\sum_{i=1}^{n-1}C_{i}{*V}_{i}}{Q_{t}}*100\%$$
where “A” is the absorbance of the samples; “C” is the concentration of drug; “Cn” is the drug concentration of the dissolution medium at each sampling time point; “Ci” is the drug concentration in the sample; “Vn” and “Vi” are the dissolution medium volume and sample volume respectively; “Qt” is the theoretical drug amount.

### 2.8. In Vitro Antioxidation: Scavenging DPPH Free Radical (DPPH·)

The scavenging effect of TMPZ on DPPH· was determined by colorimetric [20]. Prepared DPPH solution with the concentration of 0.08 mg/mL in the absolute ethanol, the experiment was divided into three groups. The blank group was set as 2 mL PBS mixed 1 mL DPPH; the sample groups included TMPZ, TMPZ-L, TMPZ-G and TMPZ-LG mixed with 1 mL DPPH, the concentration of sample was 0.18 mg/mL; the control group was a 2 mL sample mixed with 1 mL PBS. Each group was mixed evenly and left to react for 2 h in the dark. The absorbance was measured at 517 nm, and the scavenging effect (SE%) of DPPH· was calculated according to the following equation: (4)$$\mathrm{SE}\%=\frac{A_{o}-\left(A_{s}-A_{c}\right)}{A_{o}}*100\%$$
where “Ao” is the absorbance value of blank group; “As” is the absorbance value of sample groups; “Ac” is the absorbance value of control group.

### 2.9. In Vitro Antioxidation: Scavenging H_2_O_2_

It was also possible to measure the antioxidant activity of the drug in vitro by reacting it with H_2_O_2_ [21,22]. A solution of 0.4% H_2_O_2_ was prepared, and three groups were set. The blank group contained 0.6 mL of PBS mixed with 1.8 mL H_2_O_2_; sample groups included different concentration of TMPZ and TMPZ-L, TMPZ-G and TMPZ-LG with a concentration of 0.09 mg/mL, and the sample groups contained 0.6 mL of sample mixed with 1.8 mL H_2_O_2_; the control group contained 0.6 mL of sample mixed with 1.8 mL PBS; they were evenly mixed and after 10 min, the absorbance was measured at the wavelength of 230 nm, and the SE% was calculated according to Equation (4).

### 2.10. Ex Vivo Antioxidant: Inhibitory Effect on MDA Production from Liver Homogenate

We measured the inhibition rate of MDA production according to the experimental method reported by AlKreathy et al. [23]. The liver homogenate of mice was prepared first, then the mouse liver was removed, rinsed repeatedly in cold saline at 4 °C, blood stains were washed away and then it was blotted dry with filter paper, followed by the addition of 9 times the weight of cold saline (added in three parts), and then the liver was crushed with a homogenizer. The product was then centrifuged at 4000 rpm for 15 min, and the supernatant was obtained to make 10% liver homogenate.

Three groups were set; the sample groups included 100 μL sample (0.05 mg/mL) and 100 μL FeSO_4_ and 1 mL of 10% liver homogenate. The samples were a different concentration of TMPZ and TMPZ-L, TMPZ-G, and TMPZ-LG, with a concentration of 0.05 mg/mL, the positive model group included 100 μL normal saline and 100 μL FeSO_4_ and 1 mL of 10% liver homogenate; the negative blank group included 200 μL normal saline and 1 mL of 10% liver homogenate. After mixing, the groups were placed in a constant temperature shaker at 37°C for 1.5 h. Then 3 mL of thiobarbituric acid (TBA) working solution (0.375% TBA:5.6% TCA; 2:1) was added, they were placed in a water bath at 95 °C for 40 min, then cooled with running water and centrifuged at 4000 rpm for 8 min, the supernatant was sucked and its absorbance value was measured at 532 nm. The inhibition rate (IR%) was calculated according to the following Equation (5):(5)$$\mathrm{IR}\%=\frac{A_{m}-A_{s}}{A_{m}-A_{b}}*100\%$$
where “Am” is the absorbance value of positive model group; “As” is the absorbance value of sample groups; “Ab” is the absorbance value of negative blank group.

### 2.11. Ex Vivo Antioxidant: Inhibitory Effect on MDA Production Due to Low Density Lipoprotein (LDL) Being Oxidized

LDL extracts were prepared as follows: fresh blood from mice was removed, allowed to clot naturally, and centrifuged at 1000 rpm for 10 min to obtain serum. 

A 1 mL heparin citrate buffer solution (0.064 mol/L trisodium citrate was prepared with 20 mL of 5 mol/L HCL and 10 mg heparin was added to the extacts, and the pH value was adjusted to 5.04 every 100 µL of serum. After mixing, the precipitate was allowed to stand at room temperature for 10 min, centrifuged at 1000 rpm for 10 min, and the final pH was adjusted to 5.1. The precipitate was collected and weighed, and the precipitate was dissolved in 1–2 times the volume of serum in high-salt phosphate buffer (pH 7.4) and dialyzed at 4 °C for 24 h to obtain LDL extract. The FeSO_4_ solution with a concentration of 10 mmol/mL was prepared by distilled water. 

The sample groups were 1 mL of sample and 0.2 mL of FeSO_4_ and 1 mL of LDL extract, which included different concentrations of TMPZ and TMPZ-L/TMPZ-G/TMPZ-LG with the concentration of 0.0125 mg/mL. The model group was 1 mL of PBS and 0.2 mL of FeSO_4_ and 1 mL of LDL extract; the blank group was 1.2 mL of PBS and 1 mL of LDL extract. The three groups were incubated at 37 °C for 3 h, and 0.1 mL of EDTA-Na_2_ was added to stop the reaction. They were centrifuged at 3000 rpm for 10 min, 0.3 mL of the supernatant was obtained, and 2.5 mL of 20% TCA solution and 1.0 mL of 0.67% TBA were added, and then mixed well. The mixtures were soaked in boiling water for 30 min, cooled to room temperature, the absorbance was measured at 532 nm wavelength, and the IR% of MDA production due to LDL being oxidized was calculated according to Equation (5).

### 2.12. Preliminary Antioxidant Experiment of TMPZ-LG Applied to Aice Akin under Ultraviolet Rays

Three groups were randomly created (blank control group, model group and treatment group) with 30 healthy Kunming mice (female, 22 ± 2.0 g). The hair on the back of the mice was shaved with depilatory machine and depilatory cream, and the area of bare skin was about 2 cm × 2 cm. The blank control group was fed under normal light; the model group was smeared with 0.3 mL PBS and irradiated under a self-made ultraviolet light box, the vertical height between the ultraviolet lamp tube and the mice was 10–15 cm, and the initial irradiation time was 20 min/d, then the irradiation time was increased for 20 min every 5 days; the whole irradiation lasted for 30 days. The treatment group was treated in the same way as the model group, except that PBS was replaced with the same amount of TMPZ-LG. The skin thickness of the mice was measured with a vernier caliper: the left thumb and forefinger pinch the skin of mice′s back together with subcutaneous tissue, the caliper is then held with the right hand at the skin fold thickness 1 cm away from the pinch position of the left thumb; the actual thickness was equal to half of the measured thickness. The appearance of changes to the back skin of the mice (wrinkles, roughness, edema, erythema, ulceration) were observed [24].

### 2.13. Determination of Superoxide Dismutase (SOD) Activity in Mouse Skin

First, the mouse skin homogenate was prepared, washed repeatedly in 4 °C cold salt water, the blood stains were washed and then dried with filter paper, and 9 times the weight of cold salt water was added (in three parts), and then homogenized with a homogenizer and centrifuged at 4000 rpm for 15 min, then the supernatant was obtained to make 10% skin homogenate

The test group and control group were set. In the test groups, including the blank group, the model group, and treatment group, 1 mL of reagent 1 application solution (0.1 mL stock solution was used and distilled water was added to dilute to 1 mL), 0.05 mL of skin homogenate supernatant, 0.1 mL of reagent 2, 0.1 mL of reagent 3, 0.1 mL of reagent 4 application solution (prepared according to stock solution: diluent = 1:14) to the reagent tube of test group. In the control group, all the reagents were the same as above except; 0.05 mL of skin homogenate supernatant was replaced by 0.05 ml of distilled water. It was fully mixed with a vortex mixer and placed it in a constant temperature water bath at 37 °C for 40 minutes. Then, 2 mL of color developing agent was added to each reagent tube (75 mL of distilled water was added to the powder, and heated to 70–80 °C to dissolve it to make reagent 5, then 75 mL of distilled water was added to the powder to make reagent 6, that was then prepared with reagent 5; reagent 6: glacial acetic acid = 3:2:2), mixed well, and placed at room temperature for 10 min. The absorbance was determined at a wavelength of 550 nm.

### 2.14. Statistical Analysis

All the experiments in this paper were statistically analyzed for three independent repeated experiments. The data are expressed as mean and standard deviation (SD). The calculation results were plotted by Origin software (version 9.1). The mean value was compared by SPSS software (version 22.0), and the significance level was 0.05 (*p* < 0.05).

## 3. Results and Discussion

### 3.1. The Morphological Characteristics and Physical Properties of TMPZ-L

The zeta potential of TMPZ-L was negative with the value of −38.2 ± 3.1 mV, liposomes had the same charge and repelled each other, thus ensured that they would not stick together to form block and sediment, and existed in a uniform distribution form. The average particle size of liposomes was 116 ± 10.35 nm, which showed that the TMPZ-L was uniform. The zeta potential, particle size of TMPZ-L had hardly changed after 15 days of cold storage at 4 °C, which indicates that the TMPZ-L (Figure 2A) prepared by film dispersion method has good stability.

### 3.2. Encapsulation Rate of TMPZ-L

We used ultraviolet spectrophotometry to measure the ER% of TMPZ-L. We carried out the same operation on the blank liposomes, and determined whether it had absorbance at 295 nm. The results showed that the absorbance was negligible, which showed that the lipid and all other components in the liposomes had no absorption at 295 nm and they would not interfere with the absorbance of the drug; therefore, this method was feasible. The ER% of TMPZ-L was 73.05 ± 9.59%, the higher the ER%, the better the quality of liposomes. 

### 3.3. Quality Evaluation Results of TMPZ-LG

The viscosity and release rate of hydrogel with different percentages were investigated, and the optimal ratio was selected. We found that 1% CMC-Na kept the liquidity; 2% and 3% could maintain a semisolid state with suitable viscosities; 2% CMC-Na was more uniform and delicate; 4% and 5% had higher viscosities, which were not distributed well enough; and 5% CMC-Na had some agglomerate phenomenon. In addition, the R% in vitro of 1–5% TMPZ-G was tested; it was found that except for TMPZ, the R% of 1% and 2% TMPZ-G was better, so comprehensively, the viscosity of 2% CMC-Na was the most suitable choice for preparing TMPZ-G (Figure 2B). The TMPZ-LG (Figure 2C) prepared with 2% CMC-Na was well distributed, with a light yellow and transparent appearance. The result of pH value was 7.3 ± 0.5 and there was no irritation when it was applied to the skin of mice. The centrifugal test results showed that there was no stratification and discoloration of TMPZ-LG, which indicated that the liposome–hydrogels were relatively stable. 

### 3.4. Release Rate across Dialysis Membrane and Transdermal Experiments of TMPZ, TMPZ-L, TMPZ-G and TMPZ-LG

The drug preparation had different release behavior between dialysis membrane and mouse skin. According to the dialysis release curve (Figure 3A), it was found that TMPZ had the best release rate, followed by TMPZ-L, TMPZ-LG and the R% of TMPZ-G was the lowest. TMPZ-LG had a double effect as the drug storage and slow down the drug release, TMPZ not only had to escape from the bilayer of phospholipids but also break away from the shackles of hydrogel three-dimensional network structure, so the R% was always lower than that of the TMPZ solution in the same amount of time.

According to the experimental results of Zhao et al. [25], the transmissivity of TMPZ is in direct proportion to the concentration in the in vitro dialysis experiment and in vitro transdermal experiment. However, when the dialysis membrane was replaced by the back skin of mice to simulate the transdermal administration route of the human body, the release effect of TMPZ and its preparations were completely different from that of dialysis. The skin of human body is a complex structure, which consists of epidermis, dermis and subcutaneous tissue, and contains accessory organs such as sweat glands, sebaceous glands, blood vessels, lymphatic vessels and nerves. In the process of transdermal release (Figure 3B), the R% of TMPZ-L was always the highest, the dose of drug penetrating the skin was the largest and the ability to penetrate the skin was the strongest, because it had the structure of phospholipid bilayer [26], which was similar to cell membrane [27], phospholipid bilayer constitutes the basic skeleton of cell membrane and has certain fluidity. The horny layer of skin is the rate-limiting barrier of drug’s absorption, and liposome can strengthen the humidification and hydration of it [28]. On the basis of the principle of similarity and compatibility, liposomes can interact with the cell membrane and easily penetrate the cell membrane. The phospholipid membrane of the liposomes and the cell membrane of horny layer are fused with each other, resulting in the structure between the horny layer cells being changed, and forming a flat granular structure in the lipid bilayer; drug in the liposomes can easily enter the skin through the granular gap of the lipid bilayer and form the “drug warehouse” [29] between the epidermis layer and dermis layer of skin, continuously release drug, and then the drug flow through blood vessels to reach the target site could play a sustained therapeutic role [30]. 

Before the 48th hour, the R% of TMPZ was higher than that of TMPZ-LG because the structure of hydrogel slowed down the drug release rate, and the R% of TMPZ-LG was better than that of TMPZ after the 48th hour. Maybe the easy penetration of liposomes into skin played an important role. Unlike liposomes, TMPZ-G was not skin-permeable, and was bound by hydrogel, so the R% of it was the lowest. TMPZ-LG not only had the function of drug storage, but also had a good ability to penetrate the skin. It could be preliminarily considered that it is feasible to prepare TMPZ-LG for transdermal treatment, which lays a foundation for the further study of its transdermal therapeutic effect. 

In addition, we used origin software to fit the diffusion curve and adopted the Weibull CDF model (Figure 4). Weibull CDF model was the most suitable for processing in vitro release rate, the results showed that the fitted R^2^ values were great (Table 1).

### 3.5. In Vitro Results of Antioxidation of TMPZ, TMPZ-L, TMPZ-G and TMPZ-LG: Scavenging DPPH

1,1-diphenyl-2-picrylhydrazyl free radical (DPPH·) is stable with dark purple prismatic crystals. The scavenging DPPH· method is widely used to evaluate the antioxidant activity of antioxidant components in vitro and quantitatively determine the antioxidant capacity [31]. This method is based on the fact that DPPH· has a single electron and a strong absorption at 517 nm, and its alcohol solution is purple. Antioxidants (free radical scavengers) can pair single electrons, thus reducing the value of absorbance and fading the solution. Because this change is quantitatively related to the number of electrons it accepts, it can be measured by colorimetry (such as spectrophotometer).

The SE% of the TMPZ solution with different concentrations (0.075, 0.1, 0.125, 0.25 and 0.5 mg/mL) is shown in Figure 5A. The SE% was constantly rising with the increase in TMPZ concentration, even reaching nearly 80%, which was enough to show that TMPZ had strong antioxidant activity. We found that the SE% of TMPZ, TMPZ-L, TMPZ-G and TMPZ-LG to DPPH· was 50.50 ± 3.82%, 79.75 ± 7.06%, 16.57 ± 1.50% and 65.57 ± 4.13%, respectively (Figure 6A), which clearly showed that the SE% of TMPZ-L and TMPZ-LG to DPPH· was much higher than that of TMPZ (*p* < 0.01). However, in theory, the SE% of TMPZ-L and TMPZ-LG should be smaller than that of TMPZ with the same concentration because the drug could not’ be completely released in a short time under the double-layer membrane structure of liposomes and three-dimensional network of hydrogel. It might be that the excipients in liposomes or hydrogel also play a synergistic role with TMPZ in scavenging DPPH; and the SE% of TMPZ-G was very low, which indicated that the excipients in hydrogel had no scavenging effect, so the excipients in liposomes played an antioxidant role. 

### 3.6. In Vitro Results of Antioxidation of TMPZ, TMPZ-L, TMPZ-G and TMPZ-LG: Scavenging H_2_O_2_

There are several sources of hydrogen peroxide in the body, mainly in metabolism and in the process of the oxidative decomposition of substances. The production of free radicals can also be understood as the process of electron transfer [32,33], for example, NADH dehydrogenase and cytochrome C oxidase can transfer electrons to oxygen, thus generate various reactive oxygen species (ROS), including H_2_O_2_ [34]. H_2_O_2_ would attack life macromolecules and all kinds of cells if it were not cleared away in time, causing all kinds of damages at molecular, cell, tissue and organ level.

TMPZ solutions with five different concentrations (1.5, 2, 2.5, 3 and 3.5 mg/mL) reacted with H_2_O_2_ (Figure 5B). With the increase in TMPZ concentration, the SE% almost showed a straight-line rising trend and its SE% can reach 50%. The SE% of TMPZ-L and TMPZ-LG were 75.71 ± 6.13% and 73.06 ± 5.65%, which were higher than TMPZ (*p* < 0.001), but the SE% of TMPZ-G was 14.76 ± 1.2% which was lower than TMPZ (*p* < 0.01) (Figure 6B). Although the TMPZ was encapsulated in liposome–hydrogel, it still played a good antioxidant role.

### 3.7. Ex Vivo Results of Antioxidation of TMPZ, TMPZ-L, TMPZ-G and TMPZ-LG: Inhibit the Production of MDA in Liver Homogenate

MDA is the final product of lipid peroxidation, and it is a toxic substance to cells [35]; its content can be used as one of the indexes to examine the severity of cell damage. There are two ways to produce MDA, one is through the degradation of eicosanoids, such as arachidonic acid (AA) by enzymatic reaction; the second is produced by non-enzymatic oxidative degradation of polyunsaturated fatty acids (PUFAs) [36] (Figure 7). When the body is damaged by oxidation, such as skin photoaging, a large number of ROS accumulated in the body will act on the cell membrane, damaging the structure and function of the cell membrane, changing the permeability of the membrane and causing lipid peroxidation on the cell membrane to produce MDA [37], thus affecting the normal progress of a series of physiological and biochemical reactions. 

We selected TMPZ solutions with five different concentrations (0.025 0.0375 0.05, 0.0725 and 0.1 mg/mL) reacting with liver homogenate (Figure 5C), the lowest IR% of TMPZ on MDA was about 20%. The IR% of TMPZ was 20.77 ± 1.80%; the IR% of TMPZ-L was 18.14 ± 1.00%, which was lower than TMPZ (*p* < 0.05) (Figure 6C); the IR% of TMPZ-LG was 15.03 ± 0.90%, which was lower than TMPZ (*p* < 0.001). The reaction color of TMPZ, TMPZ-L, TMPZ-G and TMPZ-LG with MDA were lighter than that of the model group (Figure 8A), which means the IR% of sample groups were better, the amount of MDA produced was less, and the reaction with TBA was weakened. The IR% of MDA in each group was different from that in vitro, because the environment inside the body was very complicated, with various cells, tissues and organs including tissue fluid, plasma and lymph, so the antioxidant effect of various preparations was not as sensitive as that in vitro. Liposomes and hydrogel played a role in drug storage, so the IR% of TMPZ-L, TMPZ-G and TMPZ-LG on MDA were not as strong as TMPZ in short time, but the IR% of TMPZ-LG on MDA can still meet the needs of antioxidation.

### 3.8. Ex Vivo Results of antioxidation of TMPZ, TMPZ-L, TMPZ-G and TMPZ-LG: Inhibit the Production of MDA in LDL

Low density lipoprotein (LDL) is a kind of lipoprotein particle that carries cholesterol into peripheral tissue cells and can be oxidized into oxidized LDL [38,39]. There are many unsaturated fatty acids (PUFAs) in the core fatty acids of LDL, accounting for about 35–70% of the total fatty acids of LDL, which are prone to self-oxidation. There is a double bond structure of diallylmethylene in PUFAs, and the methylene group between the double bonds weakens the force of carbon–hydrogen bonds, and free radicals can easily obtain hydrogen from the double bonds, so PUFAs can easily form free radicals centered on carbon atoms. Carbon free radicals can easily react with oxygen molecules to generate peroxy radicals (Roo·), which then react with the double bonds in other PUFAs to obtain hydrogen to form hydroperoxide (RooH), and RooH undergoes intramolecular cleavage to generate MDA [40,41]. 

The IR% of TMPZ solutions with different concentrations (0.005, 0.01, 0.0125, 0.025 and 0.05 mg/mL) on MDA were within the range of 19.92–34.13% (Figure 5D). The IR% of TMPZ was 30.82 ± 2.03%; the IR% of TMPZ-L and TMPZ-LG was 26.89 ± 1.74% and 21.57 ± 1.25%, which were lower values than TMPZ (*p* < 0.05, *p* < 0.001); and the IR% of TMPZ-G was 9.12 ± 0.6%, which was lower than TMPZ (*p* < 0.001) (Figure 6D). The color results of the reaction were shown in the Figure 8B. The results showed that although TMPZ-LG had a sustained and controlled release effect, it could still prevent LDL from oxidizing to MDA and played a role in antioxidant activity. 

### 3.9. The Antioxidant Activity of the Liposomes’ and Hydrogel’ Component: Phospholipid and CMC-Na

Liposomes are composed of phospholipid and cholesterol, and the matrix of hydrogel is CMC-Na. In order to further verify whether the adjuvants of liposomes and hydrogels have antioxidant properties and a synergistic effect with TMPZ, we also carried out the above antioxidant tests in vitro and ex vivo on B-L and B-G, with the same specific operation steps and the same dilution times as the preparation. The SE% of DPPH· by B-L and B-G were 68.81 ± 5.17% and 0.96 ± 0.12%, respectively; the SE% of H_2_O_2_ by B-L and B-G were 58.34 ± 5.69% and 2.6 ± 0.21%, respectively. The IR% of MDA in the liver homogenate experiment were 14.08 ± 0.98% and 1.08 ± 0.1%, respectively; the IR% of LDL oxidation to MDA were 19.92 ± 1.67% and 1.03 ± 0.89%, which proved phospholipids and cholesterol in liposomes had an antioxidant effect, and were oxidized to produce oxidized phospholipids and oxidized cholesterol, while CMC-Na had no antioxidant effect. The result showed liposome–hydrogel could produce a synergistic antioxidant effect with TPMZ, and play a better antioxidant role.

### 3.10. The Results of TMPZ-LG Applied to Mouse Skin to Resist Oxidation

During the experiment, we found that the mice in the blank control group had smooth skin without erythema, ruptured ulcers and other symptoms, while the mice in the model group had rough skin with wrinkles, localized erythema and severe ulceration symptoms. The mice in the treatment group showed a milder degree of wrinkles, the area of erythema and ulceratum was smaller than that of the model group, and the symptoms were milder than those of the model group (Table 2), and we evaluated the skin symptoms of each group of mice (Table 3). The average skin thickness of the mice in each group was measured; the skin thickness of the blank control group was 0.52 ± 0.05 mm, and the skin thickness of the mice in the model group was 0.98 ± 0.12 mm, which was almost double the thickness as that of the mice in the blank control group (* *p* < 0.05), and was statistically significant. The skin thickness of the mice in the treatment group was 0.06 ± 0.038 mm which showed significant differences with the model group (* *p* < 0.05) and no significant differences with the blank control group (*p* > 0.05) (Figure 9).

The results showed that TMPZ-LG had preventive and therapeutic effects on mice, which were exposed to UV for a long time, it could effectively exert a better antioxidant effect and prevented the pathological skin changes. So, it was feasible to prepare TMPZ as a liposome–hydrogel to prevent or treat the oxidation of skin.

Liposomes had a synergistic effect with TMPZ, played a stronger antioxidant role and could penetrate the barrier of the skin well; hydrogel increased the adhesion to the skin and prolonged the retention time of the drug on the skin, so encapsulated TMPZ into liposome–hydrogel was a good choice for percutaneous treatment [42]. The results showed that TMPZ-LG exerted a good antioxidant effect ex vivo and in vitro and slow the release effect. It could effectively alleviate the symptoms of oxidative stress caused when UV was applied to the skin of mice. On the one hand, TMPZ-LG applied to the skin absorbed part of the energy of UV, which weakened UV ability to act on the skin. On the other hand, TMPZ-LG was better adhered to the skin due to its biological compatibility. Under the action of liposome, TMPZ more easily passed through the cell membrane to reach the sites where ROS gathered, inhibited the production of inflammatory factors, played an antioxidation and anti-inflammatory role and prevented the skin from producing photoaging symptoms such as erythema and wrinkles. 

### 3.11. The Results of Determination of SOD Activity in Mouse Skin

According to the experimental results, after adding the TMPZ-LG mouse skin homogenate supernatant test group, the average activity of SOD was much higher than that of the model group, but slightly lower than that of the blank group (Figure 10). SOD can eliminate ROS in organisms, catalyze the disproportionation reaction of peroxy anion and resist and block the damage of ROS to cells.

The experimental results showed that TMPZ-LG could significantly promote the SOD activity of mouse epidermis, and could almost completely make the SOD activity reach the level of non oxidation. Therefore, it is feasible to prepare TMPZ as liposome hydrogel to treat or prevent skin oxidation.

## 4. Conclusions

In conclusion, we encapsulated ligustrazine hydrochloride in liposomes and loaded it into sodium carboxymethyl cellulose hydrogel to improve the bioavailability of ligustrazine hydrochloride and skin drug delivery efficiency. The experiment shows that ligustrazine hydrochloride lipid gel has good stability, skin adhesion and slow release effect. For in vitro and in vitro experiments, TMPZ-LG showed a strong ability to scavenge free radicals and inhibit the production of MDA in mouse liver homogenate and LDL in serum. In the mouse photoaging model, TMPZ-LG significantly promoted the activity of SOD in the mouse epidermis, which proved that TMPZ could reach the place where ROS gathered through the skin mucosa with the help of lipid gel carrier, weakened the damage of ultraviolet rays to the epidermis, alleviated the photoaging symptoms of the mouse epidermis, and could become one of the methods for treating photoaging clinically in the future. Next, we will conduct more in-depth research on TMPZ-LG, such as individual sensitivity, acceptability and side effects.

## Figures and Tables

**Figure 1 polymers-14-04778-f001:**
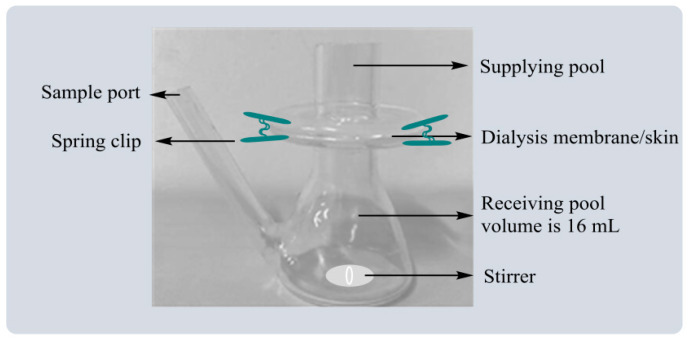
The diagram of Franz diffusion pool.

**Figure 2 polymers-14-04778-f002:**
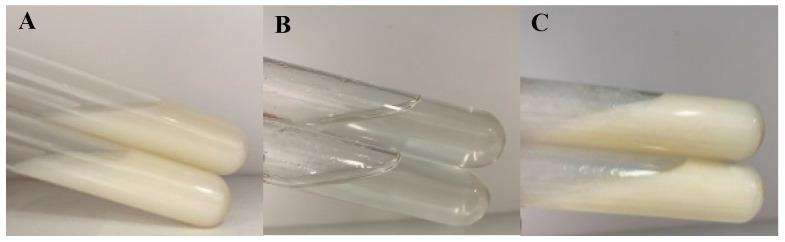
Pharmaceutical preparations (**A**) TMPZ-L; (**B**) TMPZ-G; (**C**) TMPZ-LG.

**Figure 3 polymers-14-04778-f003:**
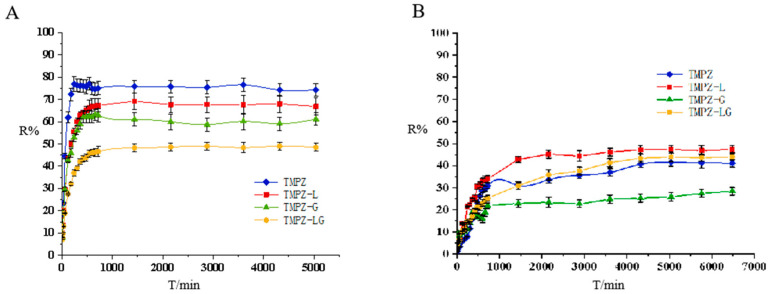
The release rate of TMPZ, TMPZ-L, TMPZ-G and TMPZ-LG. (**A**) Release rate across the dialysis membrane. (**B**) Transdermal release rate.

**Figure 4 polymers-14-04778-f004:**
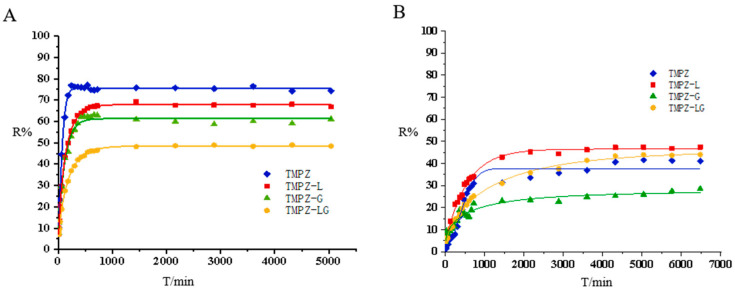
Fitting of release curve drawn by origin software. (**A**) Fitting of release curve across the dialysis membrane by Weibull CDF model. (**B**) Fitting of transdermal release curve by Weibull CDF model.

**Figure 5 polymers-14-04778-f005:**
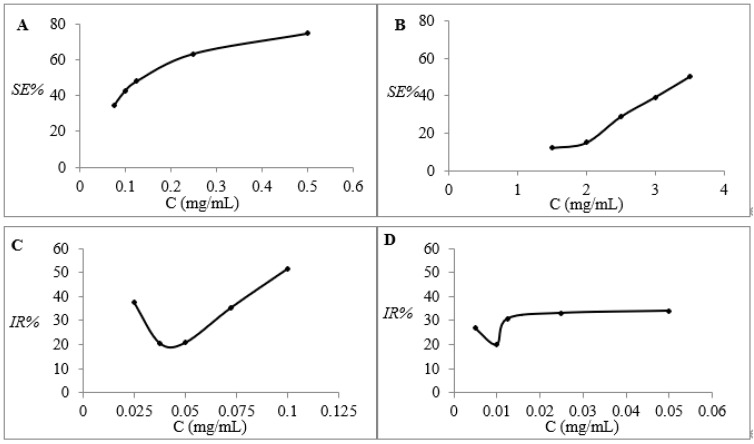
Antioxidant effects of five different concentrations of TMPZ. (**A**) Scavenging effect on DPPH. (**B**) Scavenging effect on H_2_O_2_. (**C**) Inhibiting rate on MDA production in liver homogenate. (**D**) Inhibiting rate on MDA production in low density lipoprotein.

**Figure 6 polymers-14-04778-f006:**
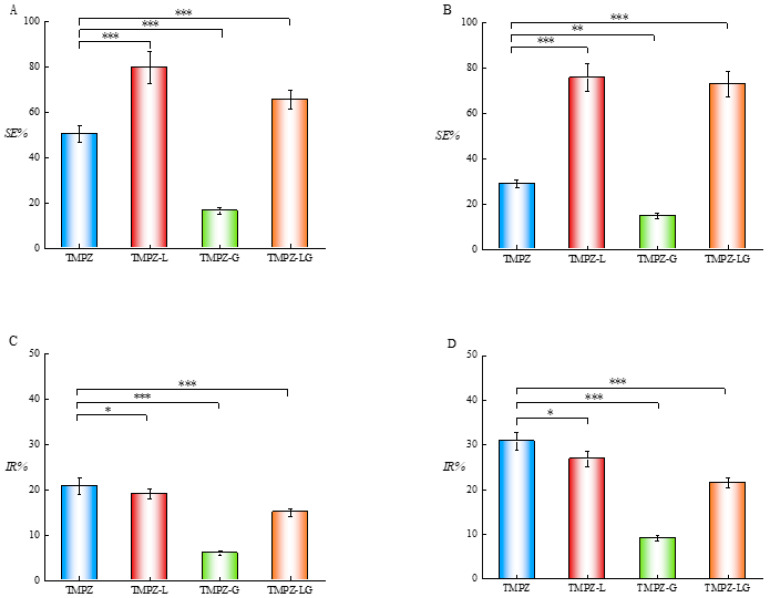
Antioxidant activity of TMPZ, TMPZ-L, TMPZ-G and TMPZ-LG. (**A**) Scavenging effect on DPPH·. (**B**) Scavenging effect on H_2_O_2_. (**C**) Inhibiting rate on MDA production in liver homogenate. (**D**) Inhibiting rate on MDA production in low density lipoprotein. *** *p* < 0.001; ** *p* < 0.01; * *p* < 0.05.

**Figure 7 polymers-14-04778-f007:**
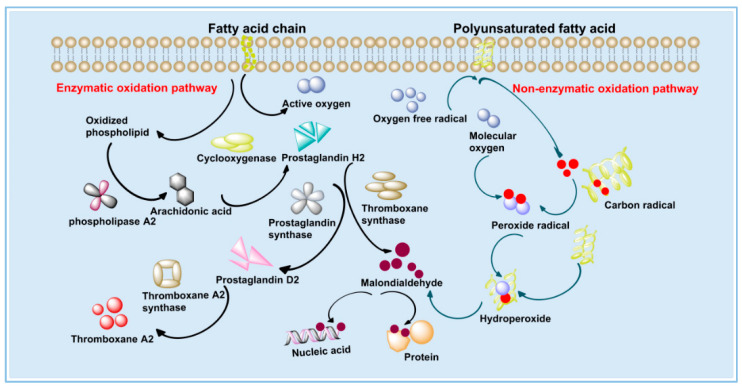
The generation route of MDA.

**Figure 8 polymers-14-04778-f008:**
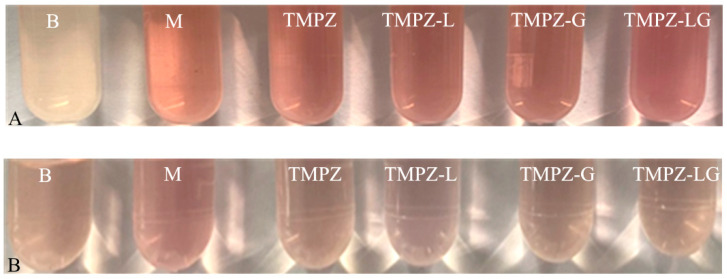
Color reaction diagram of MDA with TBA. (**A**) MDA in liver homogenate reacted with TBA; (**B**) MDA in low density lipoprotein reacted with TBA (where “B” is blank control group; “M” is model group).

**Figure 9 polymers-14-04778-f009:**
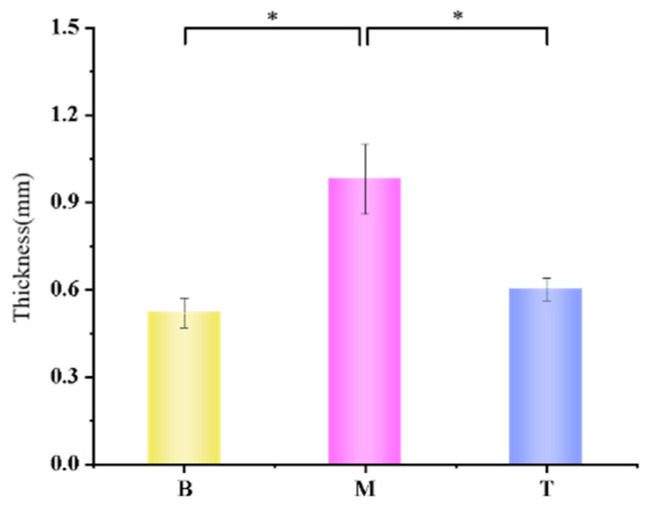
Average skin thickness of mice in each group on the 30th day (* *p* < 0.05).

**Figure 10 polymers-14-04778-f010:**
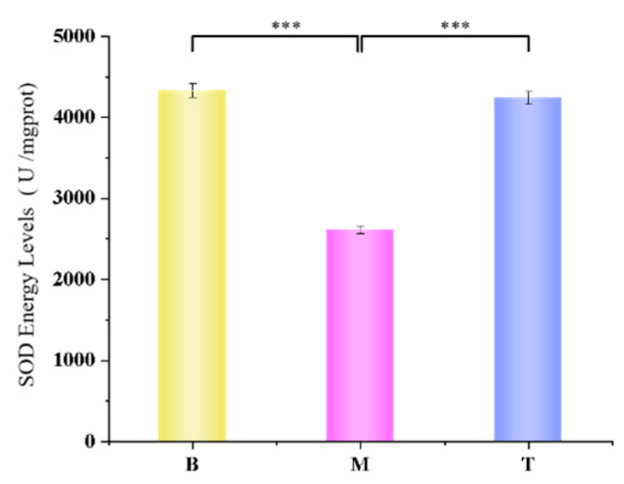
SOD activity value of mouse skin (*** *p* < 0.001).

**Table 1 polymers-14-04778-t001:** Weibull CDF model fitting of release curves for TMPZ, TMPZ-L, TMPZ-G and TMPZ-LG.

Classify	Model and Equation	Group	y_0_	A1	a	b	R^2^
In vitro release curves	Weibull CDF $y=y_{0}+A1\left(1-e^{-{\left(\frac{x}{a}\right)}^{b}}\right)$	TMPZ	8.37184	67.32706	78.34531	1.2956	0.99729
		TMPZ-L	3.40842	64.55999	130.08482	0.86904	0.99883
		TMPZ-G	6.67230	54.69415	119.81602	0.99259	0.99109
		TMPZ-LG	4.48311	44.07996	177.18407	0.86488	0.99914
Ex vivo release curves		TMPZ	2.30239	35.40483	526.85667	1.91493	0.97162
		TMPZ-L	4.66285	41.84560	557.47196	1.00453	0.99279
		TMPZ-G	3.46305	23.84930	677.53003	0.57271	0.94730
		TMPZ-LG	3.29748	42.41687	1215.72499	0.72917	0.99549

**Table 2 polymers-14-04778-t002:** Skin status of mice in each group within 30 days.

Group/Day	0	5	10	15	20	25	30
B	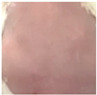	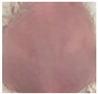	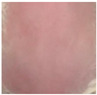	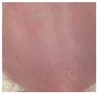	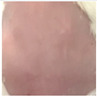	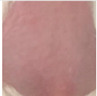	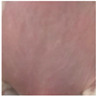
M	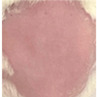	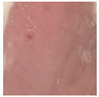	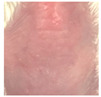	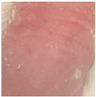	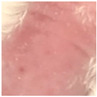	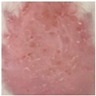	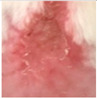
T	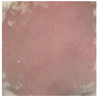	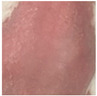	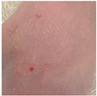	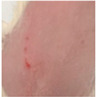	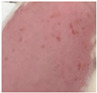	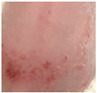	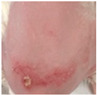

Where “B” is “blank control group”; “M” is “model group”; “T” is “treatment group”.

**Table 3 polymers-14-04778-t003:** Skin appearance evaluation standard of mice.

Group	Appearance				
	Wrinkle	Roughness	Edema	Erythema	Ulceration
B	-	-	-	-	-
M	++	+++	+++	+++	+++
T	+	+	++	++	+

Where “-” is “asymptomatic”; “+” is “mild symptoms”; “++” is “moderate symptoms”; “+++” is “severe symptoms”.

## Data Availability

Not applicable.

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
