# Peer review of "Ligustrazine as an Extract from Medicinal and Edible Plant Chuanxiong Encapsulated in Liposome–Hydrogel Exerting Antioxidant Effect on Preventing Skin Photoaging"

_polymers, 2022, doi:10.3390/polym14214778_

Round 1

Reviewer 2 Report

Ligustrazine (tetramethylpyrazine, TMP) is a bioactive ingredient extracted from the widely-used medicinal Chinese herb, Chuanxiong. It has many different properties, such as platelet aggregation inhibitors, enhancers of vessel dilation, activators of cerebral blood flow, and neuroprotective agents. The TMP compound can be developed by chemical synthesis from acetaldehyde and acetoin for industrial production in the future. 

The article's aim was to study and evaluate the antioxidant properties of TMPZ-LG on mouse skin under UV rays. It was well-written and generally understandable, However, for more scientific information, the paper significantly needs to be improved. I have some important comments and suggestions.

My comments and questions:

1. What is the gold or universal standard used in the comparison with TMPZ-LG for the skin antioxidant and anti-aging effects.? More studies such as stability, self-life, bioaccessibility, etc. should be included. 

2. The paper's Discussion is insufficient having only 240 words counted. According to the contents of the Results, the paper should be at least 1000-2000 words. Some significant points are not mentioned or discussed.

3. In the Discussion, the authors should discuss how to apply TMPZ-LG from the tissue, in-vitro and ex-vivo animal results to human practical use such as dose, standard drug and placebo, skin absorption, acceptability, individual allergy susceptibility, and other side effects, etc.? 

3. The authors may discuss the cost and benefits of the herbal cream-derived product. Commercially, what are the costs of TMP and its developed finished product?
